# Amenity Grasses—A Short Insight into Species, Their Applications and Functions

Barbara Wiewióra * and Grzegorz Żurek 

Plant Breeding & Acclimatization Institute, National Research Institute, Radzików, 05-870 Błonie, Poland
* Correspondence: b.wiewiora@ihar.edu.pl

**Abstract:** This article presents perennial grasses, without whose presence it is impossible to imagine the natural environment as well as agriculture, recreation, sport, and satisfactory aesthetics of the environment. Grasses have by far the widest distribution of all flowering families, grow on every continent, and are part of all the major biomes of the terrestrial world. They not only occur in almost all types of natural landscapes but also find a prominent place in the agricultural landscape. Grasses are not only a source of food for people (wheat, rice, maize, millet, etc.) and feed for livestock, but also a source of energy, building materials, a component of paper pulp, etc. Moreover, grasses have numerous uses to enhance the beauty of the surrounding landscape, bring relaxation, health, and comfort to people (i.e., gardens, parks, and sports facilities), and support land protection. This article describes just these, not often mentioned, and characterized grass uses, with an emphasis on the relationship between different species of perennial grasses and their functionality. The aim is to show the various aspects of the amenity use of grasses in the context of species diversity and their future under the conditions of a changing climate.

**Keywords:** turf; lawn; ornamental

## 1. Introduction

Grasses (*Poaceae*) make up one of the largest families of flowering plants. The systematic groups within the family are still changing, but between 7500 and more than 11,000 species are recognized, depending on the authority, and divided among 600–700 genera [1,2].

Wild grass communities, cultivated grasslands, and grass covers occupy 31–43% of the earth's surface [3], but in some countries, their share of the total area is much higher (Figure 1). Despite this range of estimates reflecting differences in defining grass cultivation by different authorities, grasses occupy more of the Earth's surface than any other major type of vegetation cover, i.e., forests or agriculture [4,5]. Moreover, it has been estimated that 45% of arable land is covered also by grass crops, such as wheat, maize, rice, barley, and sugarcane [6]. According to Lindner et al. [5], grasses have a significant impact on global ecology, biogeochemical cycles, and the subsistence of mankind.

Grasses co-evolved with herbivores during the Miocene epoch, spreading from forests into open habitats [5,7–10]. The 'success' of grasses is best understood in the context of their capacity to colonize, persist and transform environments—it is also called the 'Viking syndrome.' The 'Viking' success is due to efficient dispersal, fast population growth, resilience to disturbances, phenotypic plasticity, and the ability to transform environments for invaders' benefits [5].

It is postulated that the evolution of grasses and grasslands may have partially directed the evolution of humans (i.e., upright gait, the tool-using hand, and the intellect required to survive in such a diverse and demanding habitat) [11,12]. Whether or not the contribution of grasses to prehistoric human evolution is still debatable, perennial grasses are an undisputed integral component of the evolution of modern humans and societies.

Human culture, tradition, and habits around the world have been shaped by grasses and the products made from them [13].

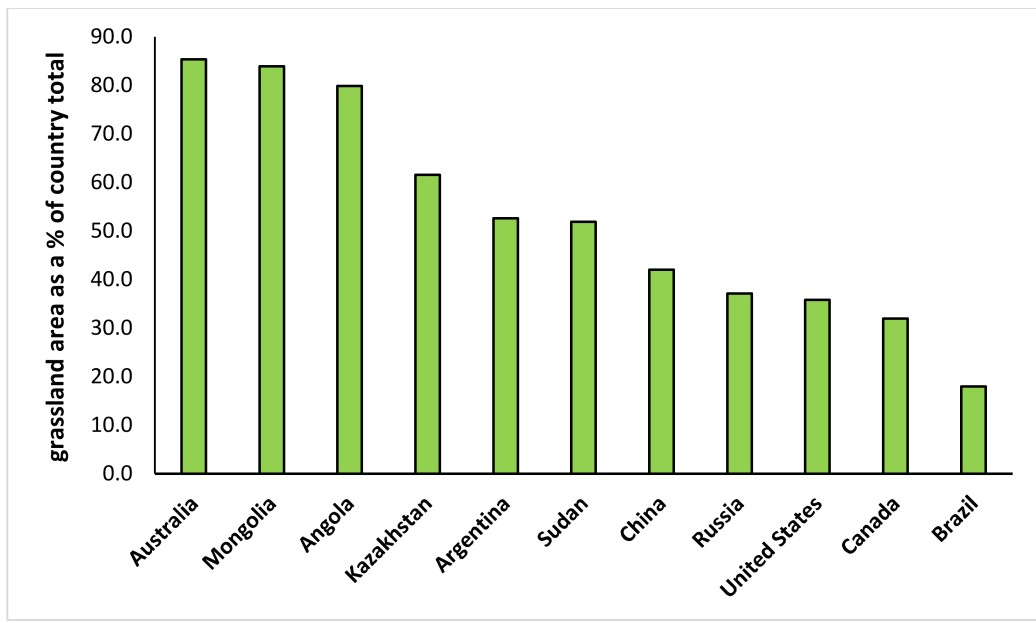

**Figure 1.** Global top countries for grassland areas (countries with >1 × 10⁶ km² of grassland) [7]; http://pdf.wri.org/page_grasslands.pdf (accessed on 1 February 2023).

However, not only have the habits and behavior of humans been 'changed' by grasses, but also those of environments. Biotic modification of environments via feedback with herbivory or fire reinforce grass dominance, leading to open ecosystems [5]. For example, the transition from woodland to grassland could be due to increased fires, which tend to benefit grasses over woody plants [14].

The evolutionary success of grasses has an impact on their current position as useful plants. This is reflected in the multitude of applications, from animal feed or sources of heat to highly specialized applications in industry, surfaces of sports facilities, or effective decorative elements in the human environment.

Despite the systematic classification of grass species, this plant family can be divided according to basic pathways of photosynthetic $CO_2$ fixation, which are further referred to as $C_3$ and $C_4$. The first was originally described in higher plants in 1957 [15] and is commonly referred to as the reductive pentose phosphate cycle or the $C_3$ pathway. This photosynthetic pathway is present in most herbaceous plant species, including grasses, especially in the temperate climate zone. In general, the $C_3$ pathway is adopted to operate at optimal rates under conditions of low temperature (15–20 °C). Therefore, this group of species is also referred to as 'cool-season grasses.'

Alternate $CO_2$ assimilation pathways were identified in sugarcane (*Saccharum officinarum* L.) in 1965 [16], referred to as the photosynthetic dicarboxylic acid cycle or the $C_4$ pathway. This pathway is adopted to operate under optimal conditions of higher temperature (30–35 °C); therefore, it concerns plants growing in the warm climate zone ('warm-season grasses').

Another classification of species within the grass family can be made based on how they are used by humans. In this article we will present the range of perennial grass species, with special attention paid to the amenity issues: lawn and ornamental (Figure 2). Bamboo has been excluded from this article, due to its exceptional ability to develop wood tissues, in contrast to other grasses which are true herbaceous plants.

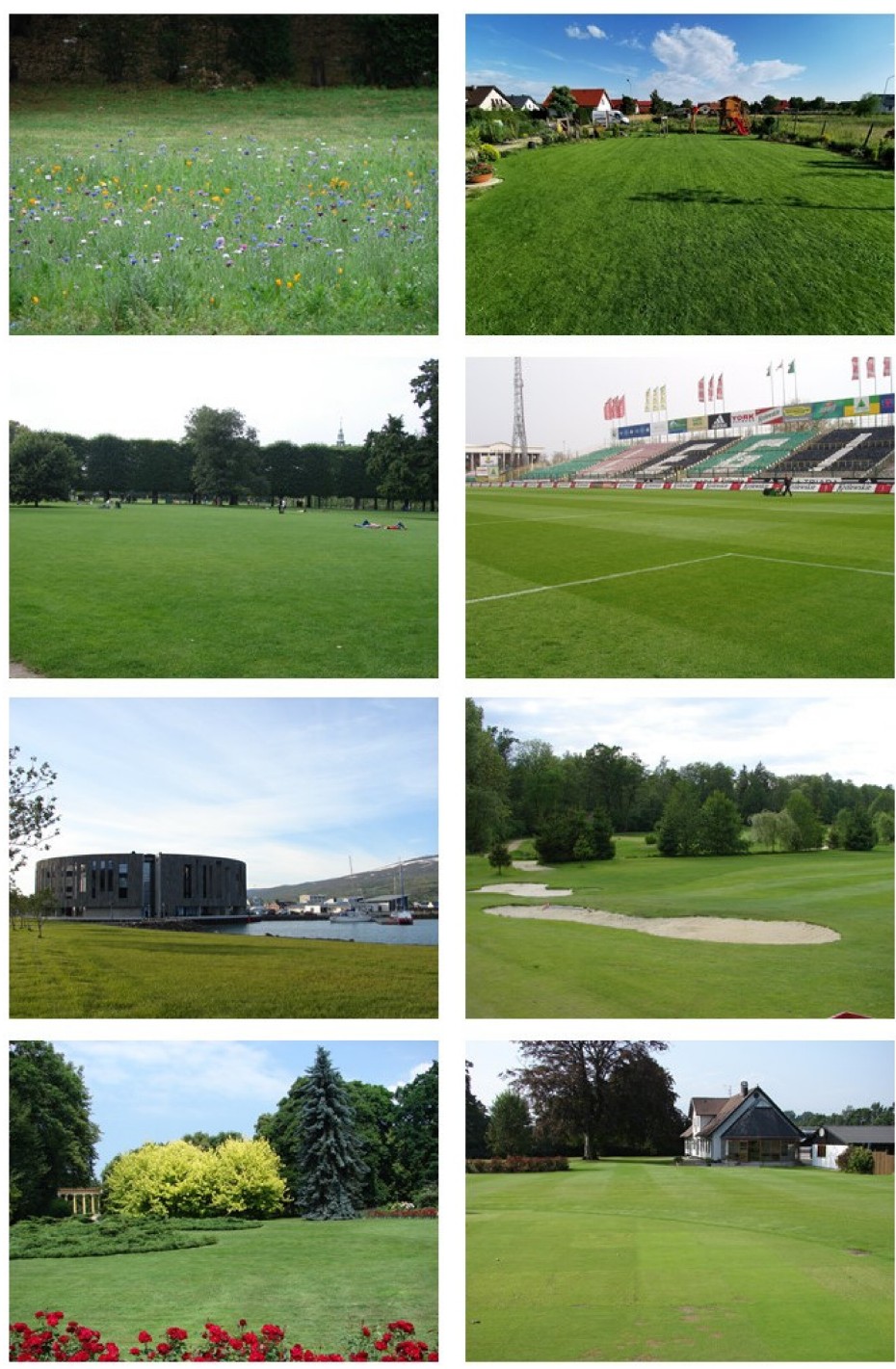

**Figure 2.** Examples of different aspects of turf grass applications. (from the left top): Flower meadow (Dallas, TX, USA); Public park (Copenhagen, Denmark); Municipal lawn (Akureyri, Iceland); Residential lawn (Zamość Castle, Poland); (from the right top): Home garden (Warsaw, Poland); Football pitch (Legia, Warsaw, Poland); Golf course (Gradi GC, Poland); Experimental golf plots (Flakkebjerg, Denmark) Author of all photographs: G. Żurek.

## 2. Functional Attributes of Grasses That Determine Their Applicability

Grasses due to so-called *phenotypic plasticity* can respond to fluctuating environments and stress by morphological and/or physiological changes [17–20]. Cultivated crops generally have reduced phenotypic plasticity compared to their wild progenitors [21–23]. Phenotypic plasticity has been defined as the change in an organism's phenotype, triggered by natural environments variation in space and time [24,25].

In other words, it is a change in the phenotype, expressed by a single genotype in different environments [21]. Phenotypic plasticity together with functional traits (i.e., physiological, morphological, or phenological features potentially affecting fitness and determining plants' response to the environment) can explain the ability of plants to grow and cope with limiting or changing resources [25,26].

For some wild grass species, including tender fountain grass (*Pennisetum setaceum* (Forsskal) Chiovenda) or common reed (*Phragmites australis* (Cavanilles) Trin. ex Steud.), high phenotypic plasticity rather than local adaptations are likely responsible for variation in invasiveness or response to windy environments [19,27]. Phenotypic plasticity and recovery differences among genotypes of big bluestem (*Andropogon gerardii* Vitm.) across a water availability gradient were postulated to allow the prediction of dominant species' responses to future drought and selection [28,29].

Grasses, like other plants, play an important role in the environment, and through this, for humans also. Considering recently developed definitions, we can call these 'ecosystem services,' which describe many different benefits to humans provided by the natural environment and healthy ecosystems [30–32]. Such ecosystems are, for example, agroecosystems, forest ecosystems, aquatic ecosystems, and grass ecosystems. Ecosystem services can be categorized into four groups [33]: supporting functions and structures; regulating services; providing services and cultural services. Of the few provisioning services, the only one dedicated to perennial grasses is where games take place (Table 1).

**Table 1.** Classification of ecosystem services [33].

| Ecosystem Services Provided by Plants Including Grasses | |
|---|---|
| **Group of Services** | **Description of Services in Groups** |
| Supportive functions and structures | • Net primary production (oxygen production via photosynthesis);<br>• Nutrient cycling, water cycling<br>• Pollinator services (herbaceous weeds in grass stands, for bees, etc.)<br>• Habitat for small fauna<br>• Biodiversity support (grassland) |
| Regulating services | • Air temperature regulation/weather amelioration<br>• Carbon sequestration<br>• Air quality<br>• Soil erosion control<br>• Water quality regulation through infiltration, regeneration of water stocks<br>• Reduction of noise pollution<br>• Reduction of urban glare and reflection |
| Provisioning services | • Plant biomass for feed, energy, materials, etc.<br>• The place for games (i.e., golf, football, cricket, etc.)<br>• Oxygen production<br>• Carbon dioxide absorption |
| Cultural services | • Enhancing private property<br>• Recreational activities (including sports, eco-tourism, etc.)<br>• Horticultural and cultural identity<br>• Social and psychological benefits |

## 3. Grasses in Amenity Applications

The majority of grass amenities are from the very beginning closely related to home lawns, which are of huge influence on people's health and behavior [34]. Considering the

definition, 'amenity' means the benefits of a property whose existence increases the value or desirability of that property. 'Amenity' in the case of grasses refers to their particular use, not related to food, forage, or bioenergy, but mostly to people's well-being. According to Cambridge Learner's Dictionary, " ... amenity means something that is intended or even *necessary* to make people's life more pleasant or comfortable" [35].

It has been well documented that natural environments and green spaces are more restorative to human physiological and psychological well-being than, for example, built environments [36,37]. Surroundings with grass have a relatively high potential to support human well-being in cities, resulting in more enthusiasm for undertaking jobs, less frustration, more patience, and fewer physical ailments [38,39]. Workers able to look at green areas (lawns, wild grasslands, trees, and ornamentals) cope better with 'directed attention fatigue' (DAF) than workers without windows to the outside [40,41].

Generally, amenity grass species can be divided into two groups: **lawn grasses**, for the creation of the green, living cover of the ground; and **ornamental grasses** for home gardening.

## 4. Lawn Grasses

Lawns originated in Medieval Europe as specially designed and artificially created grass-dominant plant communities that needed to be properly cared for to fulfill their dedicated functions [42]. In the early stages of their use (XVI-XVII centuries), the purpose of lawns was intangible, presenting symbols of power, order, and control over nature [43]. Along with technological progress, mostly with the invention of mowers, lawn management required less labor and therefore lawns became more popular and widespread in private residences [44–46].

It has been recently estimated [43,47] that, on a world scale, lawns occupy more than half of urban green areas, such as green carpets for representative buildings, decorative elements in parks, natural covers of stadiums, and playgrounds. Assuming lawns would cover 23% of cities globally, they would occupy 0.15 million to 0.80 million km$^2$ (depending on urban definitions), that is, an area bigger than England and Spain combined, or about 1.4% of the global grassland area [43]. For example, in the United States, it is estimated that lawns cover three times more area than any other irrigated crop [48].

The lawn grasses group consists of ca. 30 species (Table 2), but only a limited number of species are frequently selected for lawns, and specialized varieties are mainly created for these species in the process of breeding [49].

The most frequently used grass species in lawn applications from **cool season** grasses are perennial ryegrass (*Lolium perenne* L.), red fescue (*Festuca rubra* L.), tall fescue (*Festuca arundinacea* Schreb.), and smooth-stalked meadow grass (*Poa pratensis* L.) [43,50–53]. The dominant **warm-season** lawn grass species are bermudagrass (*Cynodon dactylon* (L.) Pers.), seashore paspalum (*Paspalum vaginatum* Swartz), St. Augustinegrass [*Stenotaphrum secundatum* (Walt.) Kunze], and zoysiagrass (*Zoysia japonica* Steudel) [43,54–61].

According to the recent edition of the OECD List of Varieties [62], the total number of registered varieties of lawn grasses exceeds 1700 in 22 species of both C3 and C4 photosynthetic pathways. This accounts for 34.9% of all grass varieties listed on the OECD list. The highest numbers of varieties belong to the most popular species as perennial ryegrass (601 lawn varieties, 37.3% of all listed varieties in this species), red fescue (311, 75.1%), tall fescue (286, 49.1%), and smooth-stalked meadow grass (228, 79.7%) (Table S1).

From a practical point of view, if grass varieties are commercially prepared for lawn applications, grass species are joined together in so-called seed mixtures. The necessity to include in the commercial offer a mixture of seeds of different species results, first of all, from the difficulties in determining the local environmental factors (soil, water, microclimate, management, etc.) present at potential customers. It is important to note that there are as many combinations of local environmental factors as there are potential customers. Many other benefits of using mixtures have been also described: higher shoot density, enhanced disease and insect resistance, better shade tolerance, earlier spring green-up, better tolerance to low maintenance, and better turf quality [50,63–69].

**Table 2.** Lawn grass species with the recommended application.

| Species | Sports Facilities | | | | | | Lawns | |
| | Golf | | | Other Ball Games | | | | |
| | Putting Greens | Tees | Fairway | Football Pitches | Tennis/ Cricket | Polo/ Racing | Residential and Public | Landscape and Low Maintenance |
|---|---|---|---|---|---|---|---|---|
| **Cool-Season Grasses** | | | | | | | | |
| *Agropyron criststum* | | | | | | | | x |
| *Agrostis canina* | xx | xx | x | | | | | |
| *Agrostis capillaris* | xx | | | | x | x | xx | x |
| *Agrostis stolonifera* | xxx | x | x | | | | | |
| *Anthoxanthum odoratum* | | | | | | | | x |
| *Brachypodium pinnatum* | | | | | | | | x |
| *Cynosurus cristatus* | | | | x | | | | x |
| *Deschampsia cespitosa* | | | | x | | | x | x |
| *Elymu glaucus* | | | | | | | | x |
| *Elymus virginicus* | | | | | | | | x |
| *Festuca arundinacea* | | | | xx | | x | x | |
| *Festuca ovina* | | | | | | | x | x |
| *Festuca rubra* sl. | xx | | | | x | x | x | x |
| *Koeleria macrantha* | | | x | | | | x | x |
| *Lolium multiflorum* | | | | x | | | | |
| *Lolium perenne* | x | x | x | xx | xx | xx | xx | x |
| *Phleum bertolonii* | | | | | | | x | x |
| *Poa arachnifera* | | | | | | | x | |
| *Poa compressa* | | | | | | | | x |
| *Poa nemoralis* | | | | | | | x | x |
| *Poa pratensis* | | x | x | xxx | | x | x | x |
| *Poa supina* | x | | | xx | | | | |
| *Poa trivialis* | x | | | | | | x | x |
| *Puccinelia distans* | | | | | | | | x |
| **Warm-Season Grasses** | | | | | | | | |
| *Bouteloa curtipendula* | | | | | | | | |
| *Bouteloa gracilis* | | | | | | | x | x |
| *Buchloe dactyloides* | | | x | | | | x | x |
| *Cynodon* sp. | xx | x | x | x | x | | x | |
| *Digitaria didactyla* | | | | | | | | x |
| *Distichlis spicata* | | | | | | | | x |
| *Eragrostis intermedia* | | | | | | | | x |
| *Eragrostis trichophora* | | | | | | | | x |
| *Eremochloa opiuroides* | | | | | | | x | x |
| *Panicum coloratum* | | | | | | | | x |
| *Panicum laxum* | | | | | | | | x |
| *Paspalum notatum* | | | | | | | x | x |
| *Paspalum vaginatum* | x | x | x | x | x | | x | x |
| *Pennisetum clandestinum* | | | | | | | x | x |
| *Stenotaphrum secundatum* | | | | | | | x | x |
| *Zoysia* sp. | | x | x | x | | | x | x |

Description: xxx—excellent suitability, xx—good suitability, x—possible to be used.

For high-maintenance lawn surfaces such as golf greens or football pitches, in many cases, few, high-quality commercial varieties of one species are mixed [70,71]. The higher the users' expectations of the turf quality and performance during exploitation, the higher the potential quality of commercial varieties selected for the mixture ought to be [72–75].

According to the UEFA recommendations for a football pitch, species used regularly on football pitches include perennial ryegrass, Kentucky bluegrass, and tall fescue for cooler regions and bermudagrass (*Cynodon* sp.), zoysia species (mainly *Zoysia japonica* Steud. And *Zoysia matrella* (L.) Merill), and seashore paspalum (*Paspalum vaginatum* O.Swartz) for warm-climate regions [76].

In some lawn grass species, the presence of specific symbiotic mutualistic relationships with fungi of genus *Epichloë* exists. These fungi, also called endophytes, live their whole life in the tissues of their hosts without visible symptoms and most of them produce alkaloid toxins in infected plants [77]. The way grasses are used can determine the nature of coexistence with fungi, and whether these will be positive or negative aspects. Endophytes' presence may bring many benefits to grasses, for example, inducing mechanisms of drought tolerance and regeneration of damage after a prolonged drought, as well as contributing to economical nitrogen management and better phosphorus digestibility [78]. In addition, grasses inhabited by endophytes are resistant to pests, nematodes, and some diseases [79].

*Epichloë* (*Neothypodium*) species of endophytes are often present on *Lolium* spp., *Festuca* spp., *Poa* spp., *Agrostis* spp., and fungus species. The most often identified in these lawn grasses are *Epichloë festucae* (*Neothypodium lolii*), *E. typhina* (*Sphaeria typhina*), *E. coenophiala* (*N. coenophialum*), or *E. uncinata* (*N. uncinatum*) and their subspecies [80].

However, endophytes do not always increase host plant resistance which was confirmed in many experiments [81–84]. Under European climatic and soil conditions, the beneficial effects of endophytes in grasses compared to plants without these fungi also are unclear [85,86]. Studies on abiotic factors such as water deficit, mowing, shading, low nitrogen fertilization, or low soil pH have demonstrated varying plant responses, depending on the plant genome—endophyte genotype structure [87–89].

The breeding programs are aimed at introducing useful endophytes into the cultivars and populations of perennial ryegrass and fescue. Existing inoculation techniques are being appropriately refined and modified for the rapid development of endophyte-containing strains and for the transfer of indicated endophytes between species [90–92]. Commercial uses of endophytes are known, and currently, endophyte presence is a benefit for lawn variety, the same as fast establishment, excellent drought tolerance, and rapid tillering [93]. However, this technology still faces great difficulties due to occasional poor levels of transmission to the seed [94,95] and the endophyte in the stored seed having a shorter life span than the seed itself [96,97] and requires further research.

*Beneficial Aspects of Turf*

Grasses in non-agricultural applications, i.e., amenity grasses, are mainly grass surfaces, hereafter referred to as turf, which provides direct contact between people and the beauty of nature. Furthermore, apart from aesthetic issues, grass surfaces perform many useful functions already described as ecosystem services. The benefits, specially dedicated to the turf, are listed below [40,98].

*Soil erosion reduction and dust stabilization.* After heavy rain, which lasted 30 min, giving ca. 75 $L \cdot m^{-2}$ of water, soil covered with grass (i.e., tall fescue in this example) lost only 10–62 $kg \cdot ha^{-1}$, while for the bare ground, it was ca. 222 $kg \cdot ha^{-1}$. From the point of view of the operation of aircraft engines, it is also very important to reduce dust near runways, maneuvering, and parking areas [98].

*Carbon sequestration.* It has been estimated that carbon storage in turf areas is comparable to that in agricultural ecosystems, although less than in forests. The annual increase of carbon in grassland on sandy loam soil was estimated at 0.52 $t \, C \cdot ha^{-1}$ [99]. A wide range of carbon accumulation rates was reported for lawns [100–102]. It depends on many factors as the age of the lawn, lawn use, fertilization, and irrigation rates, and may range from 0.22 to

3.55 t Cof water, soil covered with grassha$^{-1}\cdot$y$^{-1}$. Lawns are generally referred to as carbon sinks, except for intensively managed golf course greens and athletic fields [100,101].

*Oxygen production.* A lawn with an area of approx. 230 m$^2$ produces enough oxygen to meet the annual needs of a family of four people [40].

*Water filtration.* Turf strongly reduced surface water runoff as compared to the cultivated area. Water runoff from tobacco cultivation measured for 4 weeks was ca. 6.7 mm$\cdot$ha$^{-1}$, while from turf area it was only 0.6 mm$\cdot$ha$^{-1}$. However, with the loss of water comes a loss of nutrients also. For N and P washed away from surface runoff of the mentioned tobacco it was 2.34 and 0.48 kg$\cdot$ha$^{-1}$ per 4 weeks, respectively, while for turf it was only 0.0012 and 0.002 kg$\cdot$ha$^{-1}$, respectively [98,103,104].

*Decomposition of organic chemicals.* The bacterial population always present in turf thatch, grass clippings, and soil is one of the most effective biological systems for the decomposition of residues from chemical treatments (i.e., pesticides and organic chemicals). Therefore, turf areas can serve as catchment and filtration of runoff waters polluted with Cd, Cu, Pb, Zn, and hydrocarbons that originate from industrial waste, grease, oil, fuels, etc. Therefore, not the turfgrass itself, but the microorganisms always accompanying them serve this way.

*Cooling effect.* The temperature gradient from the grass leaf surface and surrounding air may reach 3–4 °C (Żurek, unpublished data). A lawn, located on the front of a house, provides cooling as if it were a 9-ton air conditioner [40].

*Noise reduction.* Studies have shown that natural turf absorbs sound much better than hard surfaces [98].

*Reducing the number of unwanted animals, allergens, and pathogens.* Care of backyard lawns contributes to reducing the number of undesirable creatures such as reptiles, amphibians, rodents, and insects. It is worth noting here that some insects can be vectors of dangerous diseases. Mowing lawns also reduces the number of plants that may appear that produce allergens contained in pollen [92,98].

*Human safety.* Natural turf helps to minimize injuries during sports activities, which may be more serious on synthetic surfaces or poorly or non-turfed soils. Values of impact absorption of good turf were twice higher than those obtained for bare ground (Policińska-Serwa and Prokopiuk, unpublished data).

*Increase investment value.* Money spent on landscaping compares favorably to other home improvements, such as pool, deck, or bath installations, providing one of the highest net returns to the owner. It has been estimated that turf and landscaping increase real estate value from 100 to 200%.

*The aesthetic value of turf.* Despite the increasing real value of the residence, as has been mentioned above, the turf may have a positive impact on its owner's mental health. This may also contribute to social harmony and improved productivity of residents [98].

## 5. Ornamental Grasses

Other amenity grass uses consider **ornamental** applications for landscape gardening (flower bed grasses) and floristry (dry flower arrangements). The selection of ornamental grasses for landscaping grows every year, but despite the time that passed, the words of Mary Plues are still relevant: "The Grasses tribe has many charms and attractions, which only need to be pointed out to secure the attention of all true lovers of nature" [105].

The most important for the user during the selection of particular species is the **aesthetic appeal**, which is a combination of colors, shapes, sizes, lines (vertical, ascendent, or arc-shaped), movement and sound (wind effects), and seasonal dynamics [13,106]. The visibility of the decorative aspect of grasses (i.e., plant shape, inflorescence length or discolored leaves or stems, etc.) can be very different in different cultivars of the same species [107–109].

Very important from the context of landscape architecture is that grasses provide continuing decorative effects in the landscapes long after the flowering plants have faded [110]. Grasses can be used in gardens with great results, ranging from monotonous compositions,

having the characteristics of calming elements, through combinations with more lush flowering plants to spectacular presentations of single species, usually very impressive, for example, giant reed (*Arundo donax* L.), switchgrass (*Panicum virgatum* L.), *Miscanthus* species, etc. In many historic gardens, grass was the matrix into which all other plants (shrubs, trees, perennials, etc.) were integrated [110].

The group of ornamental grasses is very abundant in species—according to the list from Table S2, there are more than 260 species in 79 genera. The genera represented by the greatest number of species are *Festuca* (18 species), *Stipa* (17 species), *Muhlenbergia* (12), *Pennisetum*, and *Miscanthus* (11 and 10 species, respectively). The majority of genera (51) are represented by only one or two species (Table 3).

**Table 3.** Ornamental grasses with genus listed according to their species abundance (according to Table S2).

| Genus Name | Number of Species in Genus |
| --- | --- |
| *Festuca* | 16 |
| *Muhlenbergia* | 12 |
| *Miscantus, Pennisetum* | 10 |
| *Andropogon, Calamagrostis, Saccharum, Stipa* | 8 |
| *Achnatherum, Nasella* | 7 |
| *Elymus, Eragrostis, Leymus, Sesleria* | 6 |
| *Cortaderia, Melica, Poa, Spartina* | 5 |
| *Chinochloa, Jarava* | 4 |
| *Arundo, Austrostipa, Chasmanthium, Glyceria, Panicum, Sorghastrum, Sporobolus, Themeda* | 3 |
| *Alopecurus, Ammophila, Arisitda, Botriochloa, Bouteloua, Brachypodium, Bromus, Ctenium, Deschampsia, Hesperostipa, Hierochloe, Holcus, Imperata, Koeleria, Melinis, Molinia, Phragmites, Pleuraphis, Schizachyrium, Tripsacum, Zizania* | 2 |
| *Ampelodesmus, Anemanthele, Arrhenatherum, Beckmannia, Blepharoneuron, Briza, Buchloe, Chodropetalum, Cymbopogon, Dactylis, Dichanthelium, Elytrigia, Hakonechloa, Helictotrichon, Hordeum, Merxmuellera, Phalaris, Setaria, Spodiopogon, Stenotaphrum, Thysanolaena, Tridens, Uniola, Vetiveria* | 1 |

In the case of 74 species, commercial varieties with specific features were also distinguished. The above list represents the vast majority of ornamental grass species, but it is not and probably could not be complete due to the huge number of institutions, companies, and individuals constantly involved in improving the already known species, creating new varieties, and searching for their best applications.

Despite their aesthetic values, perennial grasses are very popular due to their wide adaptability, quick establishment, and easy and cheap management [111]. Therefore, they are natural choices for almost every type of garden or other decorative areas [110]. Recently, with increasing shortages of water, grasses have become ideal candidates for municipal green areas due to their natural adaptability to a wide range of water requirements. Very often, moisture available on constructed or modified landscapes is far less than that of natural habitats. In such cases, it is much easier to select plant species (i.e., also grasses) naturally adapted to dry environments than install artificial irrigation. On the other hand, properly selected grass species are also very good for excess moisture in the garden.

## 6. Future of Amenity Grasses

Currently, we are observing progressing climate changes related to and caused mainly by destructive human activity. This forces us to face future expectations and regulations related to environmental protection. In the following, we will present the most frequently identified problems and possible solutions to them.

### 6.1. Reduced or Banned Pesticide and Insecticide Use

On 22 June 2022, the European Commission (EC) introduced new rules that ban all pesticides in "public parks or gardens, playgrounds, recreation or sports grounds, public paths, as well as ecologically sensitive areas". This is mostly due to protecting human health and the environment; therefore, the use of pesticides in 'sensitive areas' and within 3 m of such areas should be prohibited. Until recently, the resistance of ornamental grasses, grass used on golf courses and sports fields, to biotic and abiotic stresses was influenced mainly by appropriate management practices. However, more research is being carried out on the genetic background or the use of endophytic fungi in grass breeding, which can help increase the resistance to stress [49]. The breeding of disease-resistant grasses, in particular, is a constant race against rapidly changing pathogens. Obtaining a resistant variety is effective until a new strain of the pathogen appears, which dismisses the efforts made so far.

However, even many years of research often do not give satisfactory effects of increasing the resistance of plants, as reported by Heijden and Roulund [112], discussing the lack of significant improvement of resistance in perennial ryegrass to the red thread (*Laetisaria fuciformis*).

### 6.2. Climate Warming—Reduced $CO_2$ Emission during Turf Management

In future actions, a focus on reducing expenditures on grasses [112] should be expected in Europe, mainly by reducing fertilization or the use of protection products. It is known that mowing is the most energy-intensive cultural practice in turf management [113]. In addition, using mower fossil-fuel-driven causes the introduction of carbon mono- and di-oxides, sulfur trioxide, and hydrocarbons (HC) and their derivatives into the atmosphere.

Priest et al. [114] reported on the recent introduction of legislation, e.g., in the United States, regulating exhaust emissions from lawn and garden care devices. At the same time, they indicated that in the region of Australia they studied, lawnmowers were responsible for 5.2 and 11.6% of CO and NMHC emissions, respectively, compared to other transport sources. Reducing the frequency of mowing has a positive effect on the ecological and economic effects, because mowing, along with the use of chemicals and irrigation, are the main factors in the cost of maintaining the turf [115,116].

Research by Selhorst and Lal [116] showed that the fuel used for mowing, depending on its type, is responsible for 0.85 (for gasoline) to 0.94 kg (for diesel oil) of carbon equivalent (Ce) emitted to the atmosphere. Moreover, frequently mowed sites annually emit up to 2.0 kg m$^{-2}$ $CO_2$ [117].

Currently, alternative, more energy-efficient mowers (battery or electricity-powered) have become more popular and may reduce fuel emissions [101].

Due to the possible restrictions on the use of artificial fertilizers or reduced irrigation, alternatives are being sought to maintain the turf of the expected quality.

Cost reduction by lowering input on the use of amenity grasses is not the only positive option for users [40]. This approach is also very important due to global problems related to high greenhouse gas emissions and global warming. Many practices related to grass management, i.e., fertilization, mowing, verti-cutting, irrigation, etc., also contribute to increasing these emissions. Actions aimed at limiting these practices will reduce not only the costs incurred, but also the negative effects on the environment, and amenity grasses will be increasingly perceived as bringing significant benefits to the ecosystem services provision [49].

The projected global warming may cause genetic changes in populations in which only individuals able to adapt to the changes the fastest will survive [23]. All factors of the changing climate will have direct effects on grasses, but the duration of these effects and their impact at the level of the ecosystem is still relatively unknown. In the long term, temperature changes will probably change the geographic adaptation of grass species.

### 6.3. Increasing Drought Events and Water Shortages

An important trait in the cultivation of the major amenity grasses is their resistance to drought, but it is not clear whether they exist even in large turf trails in the USA [118].

Turf grass species vary greatly in their response to drought. It is assumed that species and cultivars with a low value of ET (evapotranspiration) and a developed root system are characterized by good 'dehydration avoidance' [119–122]. Significant differences in leaf dieback and shoot regrowth between varieties within species was also found [123–125]. In Table 4, a comparison of the drought resistance of some species of lawns and grasses is presented.

**Table 4.** Comparison of differently expressed drought resistance of some turf grass species.

| Genus, Species | Drought Resistance [119] | Regrowth after Drought (%) [123] | Green Sward after Summer Drought (%) [123] |
|---|---|---|---|
| *Agropyron cristatum* | good | - | - |
| *Festuca arundinacea* | medium | - | 100 |
| *Lolium perenne* | low | 50 | 30 |
| *Poa pratensis* | low | 80 | 10 |
| *Festuca rubra* ssp. *Comutata* | low | 75 | - |
| *Festuca rubra rubra* | low | 65 | - |
| *Agrostis capillaris* | very low | 55 | - |
| *Poa trivialis* | not resistant | 40 | - |

Differences in drought resistance between species are scientifically interesting and allow species with genetically enhanced drought tolerance to be used to improve other species. An example of such activities is the hybridization between Texas bluegrass (*Poa arachnifera* Torr.) and *Poa pratensis*, where several of the obtained varieties showed increased drought tolerance [126].

One of the main factors determining the survival of drought conditions is the effective management of water resources. Therefore, given the global problems with water, it is likely that the best solution will be to look for varieties that manage water sparingly, and not necessarily varieties that are better tolerant of periodic water deficits.

Varieties with increased resistance to drought maintain proper turgor in drought conditions due to more intensive transpiration. Unfortunately, this goes hand in hand with a higher rate of transpiration under optimal weather conditions, resulting in greater evapotranspiration of water from the soil [127]. Currently, there is no information about the possibility of producing 'water-saving' amenity grass cultivars.

Increasing problems with water availability may result in the need to learn how to use municipal water, wastewater, stormwater, or other types of water unsuitable for use by people or animals in turf areas [128].

### 7. Conclusions

Grasses are a diverse group of plants that can be found on over one-third of the Earth's surface and they have a wide range of use, including fodder, lawn, reclamation, energy, recreational and aesthetic functions. This is due to specific adaptive properties, resulting in efficient dispersal, rapid population growth, resistance to disturbances, phenotypic plasticity, and the ability to transform the environment to its advantage. In addition, increasing the existing resistance of these plants to abiotic and biotic stresses by using their symbiosis with endophytes will increase their adaptability in areas with different thermal and precipitation conditions [129]. All this information indicates the legitimacy of further research on this group of plants because the knowledge of physiological or genetic changes occurring during the adaptation processes to changing environmental conditions, along with proper phenotypic selection, will allow for their use in the breeding process.

**Supplementary Materials:** The following supporting information can be downloaded at: https://www.mdpi.com/article/10.3390/agronomy13041164/s1, Table S1. List of grass species with varieties registered according to OECD [62]. Table S2. List of ornamental grass species, its origin and cultivars presence [105].

**Author Contributions:** Conceptualization: G.Ż.; formal analysis, B.W. and G.Ż.; investigation, B.W. and G.Ż.; resources, G.Ż. and B.W.; writing—original draft preparation, B.W. and G.Ż.; writing—review and editing, B.W. and G.Ż.; visualization, G.Ż.; supervision, B.W. and G.Ż.; funding acquisition, B.W. All authors have read and agreed to the published version of the manuscript.

**Funding:** This research received no external funding.

**Informed Consent Statement:** Not applicable.

**Data Availability Statement:** Not applicable.

**Conflicts of Interest:** The authors declare no conflict of interest.

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
