# Peer review of "Amenity Grasses—A Short Insight into Species, Their Applications and Functions"

_agronomy, doi:10.3390/agronomy13041164_

Round 1

Reviewer 1 Report

The manuscript reviews information about the benefits and applications of grasses. Although the paper does not have a new concept and contain well known information, I recommend accepting it with minor revisions. In the situation where turfgrasses are blamed for every problem from potable water shortages to climate change, this well and clearly written paper is an important contribution to remind us once again about the numerous benefits of grasses.

Specific remarks:

Figure 1. The vertical axis is not labeled.  Countries with the largest share of grasslands in the total area including Poland and the World ‘(according encyklopedia.pwn.pl). This is unclear. The link is not useful.

Page 5. Some plants include common names, and some don’t. I suggest using both common and scientific names.

Page 10. ‘Turf may produce oxygen to meet the needs of four person’. The sentence does not make sense. Please correct.

Page 10. Misspelling: must be cooling (not colling) effect

Page 13. Increasing drought events and water shortages. This paragraph can be more complete. There are many reliable papers that can be mentioned about drought tolerance and the differences between cultivars, besides ‘unpublished data’. Maybe it is worth it to add info about grasses that are able to grow in extreme drought and/or salinity condition such as salt grass (Distichlis spicata).

Summing up: Since the reported results are significant for applied scientists and practitioners, I recommend accepting the paper with minor revisions.

Author Response

Dear Reviewer,

Thank you for your time and all your comments, which helped to improve the value of the article.

Specific remarks:

Figure 1. The vertical axis is not labeled.  ‘Countries with the largest share of grasslands in the total area including Poland and the World ‘(according encyklopedia.pwn.pl). This is unclear. The link is not useful.

Response: This figure has been changed,

Page 5. Some plants include common names, and some don’t. I suggest using both common and scientific names.

Response: This has been corrected in plain text. In tables – only latin names (due to lack of space).

Page 10. ‘Turf may produce oxygen to meet the needs of four person’. The sentence does not make sense. Please correct.

Response: This sentence has been changed.

Page 10. Misspelling: must be cooling (not colling) effect

Response: This has been corrected.

Page 13. Increasing drought events and water shortages. This paragraph can be more complete. There are many reliable papers that can be mentioned about drought tolerance and the differences between cultivars, besides ‘unpublished data’.

Response: Mentioned paragraph has been changed according to suggestions.

Maybe it is worth it to add info about grasses that are able to grow in extreme drought and/or salinity condition such as salt grass (Distichlis spicata).

Response: This is in fact not a typical amenity grass species, but only an example of extremely drought and salinity resistance. This has no practical application as an amenity plant and is used for livestock feeding on specific areas.

 Summing up: Since the reported results are significant for applied scientists and practitioners, I recommend accepting the paper with minor revisions.

Sincerely yours,

Barbara Wiewióra and Grzegorz Å»urek

Reviewer 2 Report

Amenity grasses – a short insight into species, their applications and functions

Barbara Wiewióra and Grzegorz Å»urek

Overview

This paper attempts to show the importance of grasses when used as turf.

It is unclear why this is important and requires a review that attempts “to show the importance of grasses through their environmental and application universality as well as its future in the context of changing climate”.  This sentence doesn’t read well and needs to be rewritten especially the phrase “importance of grasses through their environmental and application universality”. Does this mean that grasses are important because they grow everywhere?

While there is no argument that grasses are of high importance this has a poor story line and does not do the topic justice. As a general summary this paper is poorly written both in style and content. It adds very little value. It would be more suitable to a gardening magazine than a scientific journal.

There are a number of sections that haven’t been commented on – this isn’t because they are fine, it is simply because the changes and story line being used needs a total rewrite. 

Specific comments and concerns

In addition to the numerous comments on the manuscript the following are of concern.

The use of the pronouns ‘we’ and ‘our’ is a lazy style of writing. As an objective review these words should be removed.

Figure 1 - This is really concerning. Firstly the URL provided doesn’t take you to any website that mentions grassland, although for a non-Polish speaking person it is difficult to navigate. Secondly, are we expected to believe that Saudi Arabia is nearly 80% covered in grassland??? See map below

The world map below suggests that Saudi Arabia is a desert.

And why label South Africa, Turkmenistan and Australia and leave all the other bars unlabelled in the figure???  What about the prairies of the Central Lowlands and High Plains of the US and Canada, the steppes from Ukraine eastward through Russia and Mongolia, and the pampas of Argentina, Uruguay, and southeastern Brazil???

The figure below taken from supplementary information provided of a paper by Piipponen et al. 2022 - https://www.ncbi.nlm.nih.gov/pmc/articles/PMC9321565/ suggests a very different (but a more correct) perspective on percentage contribution of grasslands to country land area. Note that Saudi Arabia is in the 0 to 10% category not in the 60 to 80% category.

Table 1. Features of perennial grasses related to their applications (after Casler and Duncan, 2003, modified).

This table is pointless – the paper is about grasses and in particular amenity grasses – so what is special about the traits listed in table 1 that are peculiar to grasses. Most of the information listed is common to all plants.

 Table 2. Classification of grass ecosystem services [33].

Again the point and purpose of this table are unclear. Ref 33 is generically referring to plants in general, except perhaps providing a surface for games.

Section 4 on lawn grasses and Epichloe is very poorly written and needs a complete rewrite.

Section 6 - Future of amenity grasses.

This is also poorly written both in content and style. As an example the section on “CO2 emission during turf management” overlooks the carbon sequestration that occurs below turf. And if the land is not covered in turf what is the alternative vegetation? Arguments made are shallow and in some cases poorly substantiated.

Section 7 – concluding section is inadequate and ads little in way of a summary or conclusion.  There is general confusion throughout about grasses covering a third of the planet’s land area and the use of grasses for amenity purposes.  

Author Response

Dear Reviewers,

Thank you for your time and all your comments, which helped to improve the value of the article.

Rev. # 2

Overview

This paper attempts to show the importance of grasses when used as turf.

It is unclear why this is important and requires a review that attempts “to show the importance of grasses through their environmental and application universality as well as its future in the context of changing climate”.  This sentence doesn’t read well and needs to be rewritten especially the phrase “importance of grasses through their environmental and application universality”. Does this mean that grasses are important because they grow everywhere?

While there is no argument that grasses are of high importance this has a poor story line and does not do the topic justice. As a general summary this paper is poorly written both in style and content. It adds very little value. It would be more suitable to a gardening magazine than a scientific journal.

There are a number of sections that haven’t been commented on – this isn’t because they are fine, it is simply because the changes and story line being used needs a total rewrite. 

Author response: summary has been rewritten.

Specific comments and concerns

In addition to the numerous comments on the manuscript the following are of concern.

The use of the pronouns ‘we’ and ‘our’ is a lazy style of writing. As an objective review these words should be removed.

Author response: above has been changed according to reviewer suggestions.

Rev.: These two sentences are not in agreement - so is it 31-43% or 26%??

Author response: The sentences have been corrected

Figure 1 - This is really concerning. Firstly the URL provided doesn’t take you to any website that mentions grassland, although for a non-Polish speaking person it is difficult to navigate. Secondly, are we expected to believe that Saudi Arabia is nearly 80% covered in grassland??? See map below

The world map below suggests that Saudi Arabia is a desert.

And why label South Africa, Turkmenistan and Australia and leave all the other bars unlabelled in the figure???  What about the prairies of the Central Lowlands and High Plains of the US and Canada, the steppes from Ukraine eastward through Russia and Mongolia, and the pampas of Argentina, Uruguay, and southeastern Brazil???

The figure below taken from supplementary information provided of a paper by Piipponen et al. 2022 - https://www.ncbi.nlm.nih.gov/pmc/articles/PMC9321565/ suggests a very different (but a more correct) perspective on percentage contribution of grasslands to country land area. Note that Saudi Arabia is in the 0 to 10% category not in the 60 to 80% category.

Author response: figure 1 has been changed according to other data source. You were right, some mistakes has been made in cited data source.

Section 2

Author response: All sentences indicated by the reviewer have been rewritten

Table 1. Features of perennial grasses related to their applications (after Casler and Duncan, 2003, modified).

This table is pointless – the paper is about grasses and in particular amenity grasses – so what is special about the traits listed in table 1 that are peculiar to grasses. Most of the information listed is common to all plants.

 Author response: table has been deleted.

 Table 2. Classification of grass ecosystem services [33].

Again the point and purpose of this table are unclear. Ref 33 is generically referring to plants in general, except perhaps providing a surface for games.

Author response: some minor changes has been made in this table. However, in our opinion it is valuable to show the importance of grasses (as well as other plants) in the environment and to the people.

Section 4 …on lawn grasses and Epichloë is very poorly written and needs a complete rewrite.

Author response: this section has been changed, especially in accordance to endophyte section

Section 6 - Future of amenity grasses.

This is also poorly written both in content and style. As an example the section on “CO2 emission during turf management” overlooks the carbon sequestration that occurs below turf. And if the land is not covered in turf what is the alternative vegetation? Arguments made are shallow and in some cases poorly substantiated.

Author response: carbon sequestration has been described in more detail in section 4.

Section 7 – concluding section is inadequate and ads little in way of a summary or conclusion.  There is general confusion throughout about grasses covering a third of the planet’s land area and the use of grasses for amenity purposes. 

Author response: some changes in this section has been made, some phrases has been removed.

Sincerely yours,

Barbara Wiewióra and Grzegorz Å»urek

Round 2

Reviewer 2 Report

Definitely an improved version. Very please to see a new Figure 1 – still can’t believe the original was ever included. Also great to see old table 1 removed.

Some italics missing in paragraph below

Also in the conclusion the bracket below needs to be removed.

Author Response

Thank you for your time and all the comments that increased the value of the paper.

Sincerely yours,

Barbara Wiewióra